# Particleboards from Recycled Pallets

**Ján Iždinský *, Ladislav Reinprecht and Zuzana Vidholdová**

Department of Wood Technology, Faculty of Wood Sciences and Technology, Technical University in Zvolen, T. G. Masaryka 24, 96001 Zvolen, Slovakia; reinprecht@tuzvo.sk (L.R.); zuzana.vidholdova@tuzvo.sk (Z.V.)
* Correspondence: jan.izdinsky@tuzvo.sk

**Abstract:** Worldwide production of wooden pallets continually increases, and therefore in future higher number of damaged pallets need to be recycled. One way to conveniently recycle pallets is their use for the production of particleboards (PBs). The 3-layer PBs, bonded with urea-formaldehyde (UF) resin, were prepared in laboratory conditions using particles from fresh spruce logs (FSL) and recycled spruce pallets (RSP) in mutual weight ratios of 100:0, 80:20, 50:50 and 0:100. Particles from RSP did not affect the moisture properties of PBs, i.e., the thickness swelling (TS) and water absorption (WA). The mechanical properties of PBs based on particles from RSP significantly worsened: the modulus of rupture (MOR) in bending from 14.6 MPa up to 10 MPa, the modulus of elasticity (MOE) in bending from 2616 MPa up to 2012 MPa, and the internal bond (IB) from 0.79 MPa up to 0.61 MPa. Particles from RSP had only a slight negative effect on the decay resistance of PBs to the brown-rot fungus *Serpula lacrymans*, while their presence in surfaces of PBs did not affect the growth activity of moulds at all.

**Keywords:** particleboards; pallets; recycled wood; physical and mechanical properties; decay and mould resistance; FTIR

## 1. Introduction

Today, an increasing emphasis is placed not only on the recycling of municipal waste but also on other assortments, such as wood, which can be reused for the production of wooden agglomerated materials, especially particleboards (PBs). The measures introduced by the EU seek to close the life cycle of products and materials by preserving, as far as possible, their value to the economy, minimizing waste production and maximizing recycling and reusing. Thus, the benefits are mainly in the environment and the economy [1,2].

The idea of wood waste recycling has only started to be taken more seriously in recent decades, as seen in the work of Ihnát et al. [2]; in the past, researchers have dealt with the issue of wood recycling predominantly at the theoretical and laboratory levels. However, application in industry has been progressively and over time, considering more economic [3] and environmental benefits [4–6]. Today, old waste wooden products—mainly pallets, drums and furniture from solid wood and wooden composites (PBs, oriented strand boards (OSBs), plywood, etc.)—represent a significant resource for manufacturing new PBs [2,7].

Accordingly, in the German Decree for material recycling and thermal recycling, waste wood is categorized into four classes (A I to A IV), found in the attachments of the German Waste Wood Ordinance [8]. Pallets are divided into four categories: (I) Pallets made of solid wood, (II) Pallets of composite wood materials, (III) Other pallets with composite materials, (IV) Wood pallets with preservatives and other wood with high pollutant content intended for energy use only [8].

Pallets are used for storing, protecting, and transporting freight. They are the most common base for handling and moving the unit load, carried by materials handling units,

such as forklifts [9]. The pallet market continually increases due to the rising standard of goods transportation, the adoption of modern material handling units in different industries, and market demand for palletized goods. More than 600 million Euro-pallets are in circulation in the global logistics industry. In 2020, 123.5 million wooden the European Pallet Association e.V. (EPAL) pallets and other carriers were produced, which is 0.5 million more compared to 2019 [10].

Approximately 450–500 million new pallets are manufactured annually and join approximately 2 billion pallets that are in circulation in the U.S. [11]. In the European Union, some 280 million pallets are in circulation every year [12].

The two most common pallet types in the European market are the Euro-pallet and the industrial pallet. Both pallet types are standardized and used in the European transport market. The Euro-pallet has a length of 1.2 m and a width of 0.8 m with a weight of 25 kg when made of wood. The industrial pallet length amounts to 1.2 m and its width is 1.0 m with a weight of 35 kg. The EPAL indicates the load capacity for Euro-pallets of 1500 kg and for industrial pallets of 1250 kg [13].

Pallets are commonly made of wood, which is one of the raw materials often used for containers and is the world's most important renewable material and regenerative fuel [14,15]. Due to the fact that packaging material has already fulfilled its function at the beginning of the use phase of the respective product and is then turned into waste, the environmental relevance of containers materials has become very important [16]. The production of PBs only from recycled pallets of a defined type has not been the subject of much research, because in practice, the pallets usually represent a mix of various types and are made from different wood species [17].

The aim of this work was to study the effect of recycled wood particles—recyclates from recycled spruce pallets (RSP) added into PBs together with various amounts of spruce particles prepared from fresh spruce logs (FSL)—on the selected physical, mechanical and biological properties of PBs.

## 2. Materials and Methods

### 2.1. Materials

2.1.1. Wood Particles

In the experiment, two types of wood particles prepared from the fresh spruce logs (FSL = C/control/) and the recycled spruce pallets (RSP) were used. Both wood materials were obtained and processed in the company Kronospan s.r.o., Zvolen, Slovakia (Figure 1a). Selected mechanical properties of the FSL were partly higher than those of the RSP: A) the compression strength parallel to the grain ($\sigma_{Compression\parallel}$) by STN 49 0110 [18] for 15 sample-replicates was 42.3 MPa and 38.2 MPa, respectively; B) the dynamic modulus of elasticity ($MOE_d$) measured with the ultrasonic timer Fakopp for 12 slab-replicates at their moisture content of app. 30% was 17.73 GPa and 14.51 GPa (Figure 1b), respectively.

From the FSL, chips were prepared on drum chipper TH/N 1000/1250/15 (Rudnick & Enners Maschinen- und Anlagenbau GmbH, Alpenrod, Germany). From the RSP, chips were prepared using the mobile chipper HEM (JENZ GmbH, Petershagen, Germany) combined with the magnetic separation of ferrous metals (iron, steel, etc.), i.e., nails and other connectors on pallets. Following this, wood particles were prepared from wood chips using the Knife Ring Flakers G24 (GOOS Engineering spol. s.r.o., Brno, Czech Republic).

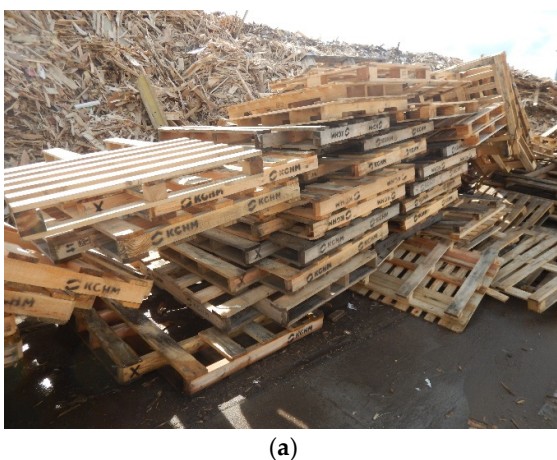 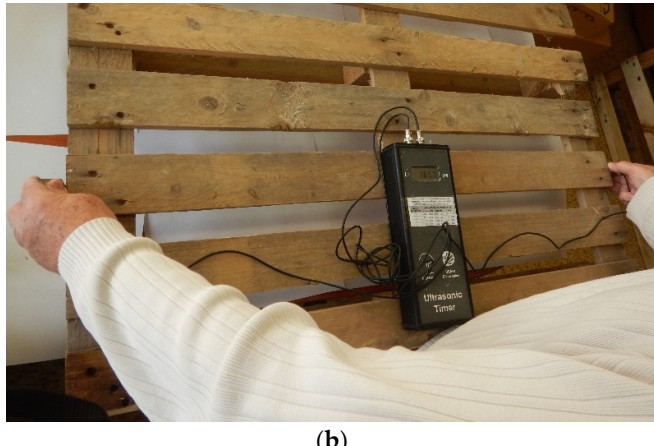

(**a**)          (**b**)

**Figure 1.** Views on the storage of discarded pallets in Kronospan s.r.o., Zvolen (**a**)**,** and on the determination of the modulus of elasticity (MOE$_d$ = density × /speed of ultrasonic waves in the longitudinal direction/$^2$) of slab in pallet by the ultrasonic timer Fakopp (**b**).

Finally, spruce wood particles were individually milled on finer particles for core and surface layers in the laboratories of the Technical University in Zvolen, using the grinding mill SU1 (TMS, Pardubice, Czech Republic). The dimensions of particles selected for the core layer of PBs were from 0.25 to 4.0 mm, and for the surface layers from 0.125 to 1.0 mm (Figure 2). Particles for the core layer were dried to a moisture content of 2%, and for the surface layers to a moisture content of 4%.

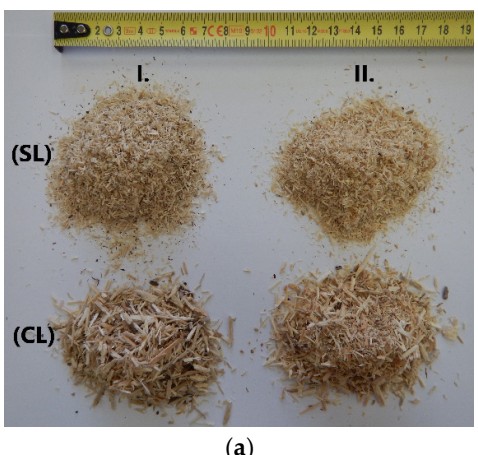 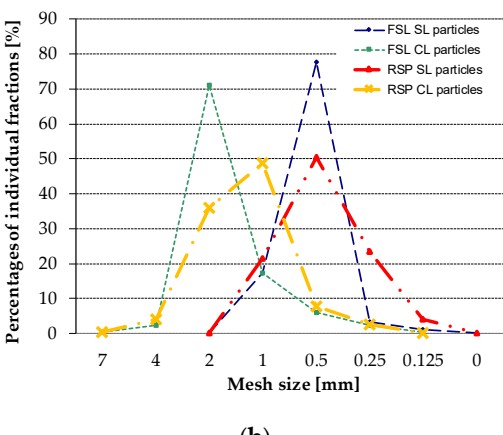

(**a**)          (**b**)

**Figure 2.** Particles for PBs: (**a**) Prepared from the fresh spruce logs (FSL)—type I., and from the recycled spruce pallets (RSP)—type II., at which both particle types were used in the surface layers (SL) and the core layer (CL) of PBs; (**b**) Size characteristics of particles from FSL and RSP, with percentage of individual fractions for surface layers (SL) and core layer (CL).

Particles with a dimension of ≤0.25 mm were subjected to FTIR spectra analysis using a Nicolet iS10 spectrometer equipped with Smart iTR ATR accessory using diamond crystal. For both particle types, 4 spectral measurements were performed in the range from 4000 cm$^{-1}$ to 650 cm$^{-1}$ with a resolution of 4 cm$^{-1}$. Measured spectra were baseline corrected and analyzed in absorbance mode by OMNIC 8.0 software (Table 1).

**Table 1.** Intensity of FTIR spectra (normalized at 898 cm⁻¹) for particles from the fresh spruce logs (FSL) and the recycled spruce pallets (RSP).

| FTIR (cm⁻¹) | FSL | RSP |
|---|---|---|
| 1274 | 1.64 | 1.24 |
| 1334 | 0.15 | 0.13 |
| 1372 | 1.11 | 1.15 |
| 1430 | 1.06 | 1.09 |
| 1510 | 2.42 | 2.28 |
| 1600 | 0.75 | 0.51 |
| 1653 | 0.17 | 0.18 |
| 1730 | 1.09 | 1.28 |
| 2900 | 1.60 | 1.63 |
| *TCI* = 1372/2900 | 0.69 | 0.71 |
| *LOI* = 1430/898 | 1.06 | 1.09 |

### 2.1.2. Resin and Additives

Two urea-formaldehyde (UF) resin types, produced in company Kronospan, were used for mixing with wood particles: (a) KRONORES CB 4005 D for the surface layers of PBs added to particles in an amount of 11 wt.%, and (b) KRONORES CB 1637 D for the core layer of PBs added to particles in an amount of 7 wt.%. The basic characteristics of these two UF resins were as follows: molar ratio of urea to formaldehyde constantly 1 to 1.2; solid content 65.9% and 67.4% by EN 827 [19]; viscosity 76 s and 86 s using Ford cup 4 mm/20 °C by EN ISO 2431 [20]; pH value 9.08 and 8.62 by EN 1245 [21]; gel time 81 s and 36 s by chloride test, respectively. The $NH_4NO_3$, used as 57 wt.% water solution, was applied as a hardener for both UF resins, added to their dry mass in amount of 2% and 4%, respectively. Paraffin, used as 35 wt.% water emulsion, was applied on the surface and core particles in amounts of 0.6% and 0.7%, respectively. A higher amount of hardener in the UF resin for the core layer of PB (4%) was needed due to technological conditions of the PBs preparation, i.e., because the temperature increase (needed for curing of UF resin) starts later in core layer as in the surface layers of PBs.

### 2.2. Particleboard Preparation

The 3-layer PBs with the dimensions of 400 mm × 300 mm × 16 mm were prepared in laboratories of the Technical University in Zvolen. The ratio of surface/core wood particles for all PB-types was constant 35:65. The moisture content of particles mixed with UF resins was 8.5–10.2% in the surface layers (SL), and 5.9–6.9% in the core layer. The PBs were prepared by the same technology as used Iždinský et al. [17,22]—i.e., firstly cold pre-pressing of particle mats at 1 MPa, followed by hot pressing in pressure (CBJ 100–11 laboratory press, TOS, Rakovník, Czech Republic) at a maximum temperature of the pressing plates in the press 240 °C, a maximum pressing pressure of 5.75 MPa, and a pressing factor of 8 s/mm (total pressing time of 128 s). From each PB type, 6 boards were produced, i.e., 24 PBs in total (Table 2).

**Table 2.** Individual types of manufactured particleboards (PBs).

| Variant of PB | Recycled Spruce Pallets (RSP) in PB *w* (Recycled Wood)/*w* (Total Wood) (%) | Number of Produced Boards | Board Type |
|---|---|---|---|
| PB-C: 100% particles from fresh spruce logs (FSL) | 0 | 6 | C |
| PB-RSP: 20%, 50% or 100% particles from recycled spruce pallets (RSP), combined with FSL particles | 20 | 6 | RSP-20 |
| | 50 | 6 | RSP-50 |
| | 100 | 6 | RSP-100 |

### 2.3. Properties of PBs-Physical, Mechanical, and Biological

Selected physical, mechanical and biological properties of PBs were determined on samples cut out from PBs (Figure 3), in accordance with European (EN) and Slovak (STN) standards.

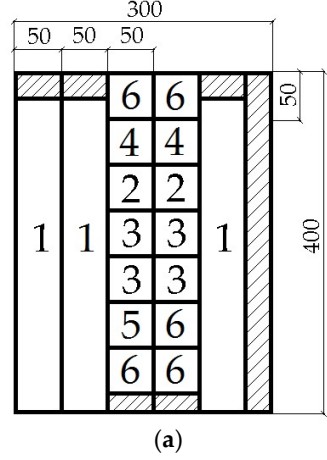

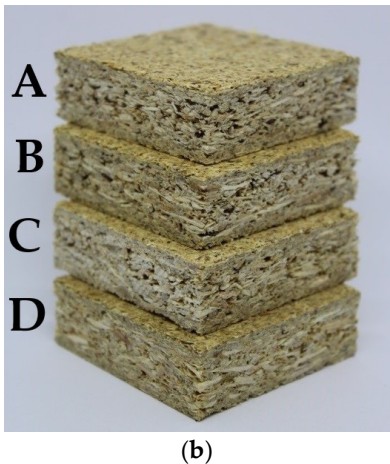

|     |     |
| --- | --- |
| (**a**) | (**b**) |

**Figure 3.** Scheme of samples preparation from the PB (**a**): 1—MOR and MOE by 3-point bending test [23], 2—TS and WA after 2 and 24 h [24,25], 3—IB [26] and density [27], 4—decay resistance to the fungus *Serpula lacrymans* [28], 5—mould resistance [29], 6—spare samples; (**b**) Display of PB-samples 50 mm × 50 mm × 16 mm with different amount of particles from recycled spruce pallets (RSP) *w/w* (%): (A) C = 0%, (B) 20RSP = 20%, (C) 50RSP = 50%, and (D) 100RSP = 100%.

Physical properties: the density by EN 323 [27], the moisture content (w) by EN 322 [30], and the thickness swelling (TS) and water absorption (WA) after 2 and 24 h by EN 317 [24] and STN 490164 [25] were determined.

Mechanical properties using the universal machine TiraTest 2200 (VEB TIW, Rauenstein, Germany): the modulus of rupture (MOR) and modulus of elasticity (MOE) in bending by EN 310 [23]; and the internal bond (IB) by EN 319 [26] were determined as well.

Biological properties: the decay resistance to the brown-rot fungus *Serpula lacrymans* (Schumacher ex Fries) S.F.Gray/*S. lacrymans* (Wulfen) J. Schröt., by IndexFungorum/, strain BAM 87 (Bundesanstalt für Materialforshung und -prüfung, Berlin) performed according to a partly modified ENV 12038 [28] (changes to [28] listed by Iždinský et al. [22])—determining after 16 weeks the mass loss (Δm) and moisture content ($w_{decayed}$) of PBs; and the mould growth activity (MGA) of the mixture of microscopic fungi (*Aspergillus versicolor* BAM 8, *Aspergillus niger* BAM 122, *Penicillium purpurogenum* BAM 24, *Stachybotrys chartarum* BAM 32 *and Rhodotorula mucilaginosa* BAM 571) on the top surface of PBs by a partly modified EN 15457 [29] (changes to [29] listed by Iždinský et al. [22])—using these criteria: 0 = without mould growth; 1 = mould up to 10%; 2 = mould up to 30%; 3 = mould up to 50%; and 4 = mould more than 50%.

### 2.4. Statistical Analyses

By the statistical software STATISTICA 12 (StatSoft, Inc., Tulsa, OK, USA) were analyzed the gathered data (the arithmetic mean and standard deviation) and stated the simple linear correlations with the coefficient of determination ($r^2$) and the probability value (*p*-value) that measures the likelihood that a test-statistic value could occur by chance, given that the null hypothesis is true.

## 3. Results and Discussion

### 3.1. Physical and Mechanical Properties of PBs

For the final moisture and strength properties of composites, including PBs, two basic factors are important: (a) structure and properties of the input materials; (b) technological

processes of the composite preparation. The moisture and strength properties of PBs are significantly influenced by the type, amount and strength of chemical bonds (etheric and other types of covalent bonds created between the wood components and the resin) and physico–chemical interactions (hydrogen bonds and van der Walls interactions created in the mixture of wood substrate and the resin) by which are in them mutually connected the individual wood particles [31,32].

The basic physical and mechanical properties of PBs are presented in Table 3. The average density of all four PB-types ranged in a very narrow interval from 651 kg·m⁻³ to 657 kg·m⁻³ (Table 3, Figure 4). The densities of the laboratory prepared PBs were comparable with the density of commercial PBs, at which they were not affected by different sources of spruce wood from which the particles for the core and surface layers were prepared.

**Table 3.** Physical and mechanical properties of the reference/control PB (PB-C) and the PBs containing particles from recycled spruce pallets (PB-RSP).

| Property of PB | | Recycled Spruce Pallets (RSP) in PB *w/w* (%) | | | |
|---|---|---|---|---|---|
| | | 0 | 20 | 50 | 100 |
| Density | [kg·m⁻³] | 656 (15.7) | 651 (18.1) | 657 (21.8) | 653 (26.0) |
| Thickness swelling (TS) after 2 h | [%] | 6.00 (0.53) | 5.27 (0.27) | 11.02 (1.63) | 6.47 (0.84) |
| Thickness swelling (TS) after 24 h | [%] | 23.81 (1.38) | 18.67 (1.58) | 27.87 (1.63) | 23.67 (1.25) |
| Water absorption (WA) after 2 h | [%] | 27.43 (2.04) | 18.49 (0.92) | 41.11 (3.62) | 20.62 (0.84) |
| Water absorption (WA) after 24 h | [%] | 68.31 (2.32) | 50.95 (2.30) | 76.80 (2.24) | 56.77 (2.82) |
| Internal bond (IB) | [MPa] | 0.79 (0.06) | 0.70 (0.05) | 0.68 (0.04) | 0.61 (0.03) |
| Modulus of rupture (MOR) | [MPa] | 14.6 (1.56) | 12.1 (0.97) | 12.4 (1.02) | 10.0 (1.27) |
| Modulus of elasticity (MOE) | [MPa] | 2616 (286) | 2471 (390) | 2276 (248) | 2012 (193) |

Notes: Mean values of density are from 42 samples, of TS from 12 samples, of WA from 12 samples, of IB from 24 samples, of MOR and MOE from 18 samples. Standard deviations are in the parentheses. The commercial PB type P2 (i.e., by EN 312 [33] boards in interior, including furniture, exposed to dry conditions) has these minimal limits of searched mechanical properties: IB = 0.35 MPa, MOR = 11 MPa, MOE = 1600 MPa.

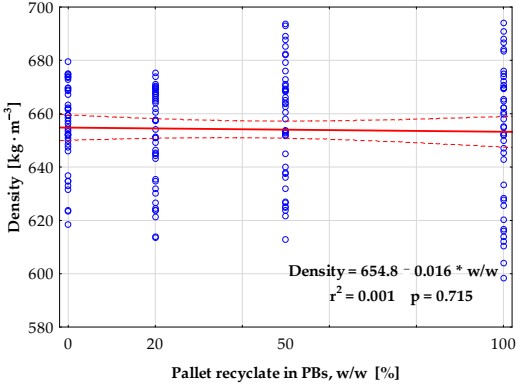

**Figure 4.** Density of PBs containing different amount of particles from recycled pallets.

The particles from RSP had only a negligible impact on the moisture properties of PBs, i.e., the TS and WA (Table 3, Figures 5 and 6). It is evident from the linear correlations "TS or WA = a + b × *w/w*" between the percentage weight content of RSP particles in PBs (*w/w*) and the TS and WA values of PBs, which were characterized by very small coefficients of determination $r^2$ from 0.004 to 0.07, and with high p-values from 0.077 to 0.664.

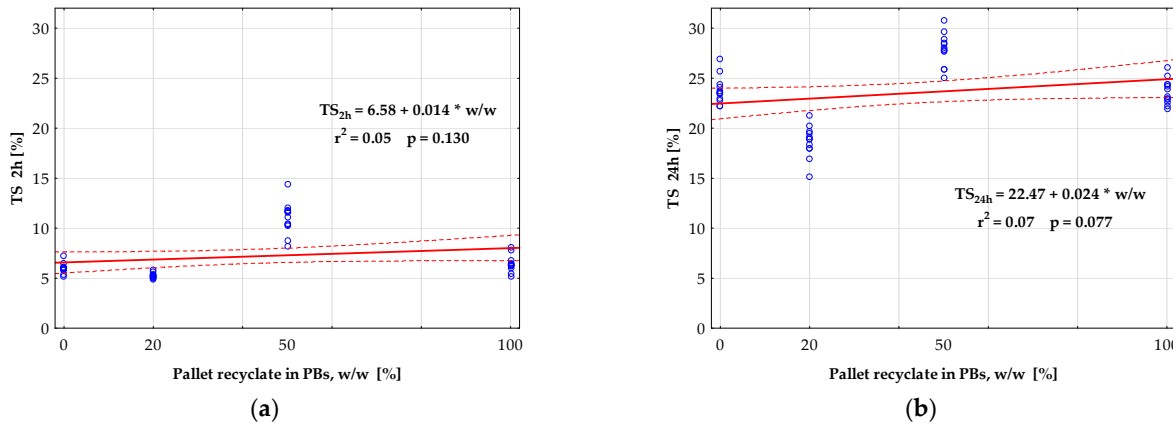

**Figure 5.** Thickness swelling (TS) after 2 h (**a**) and 24 h (**b**) of PBs containing different amount of particles from recycled spruce pallets (RSP).

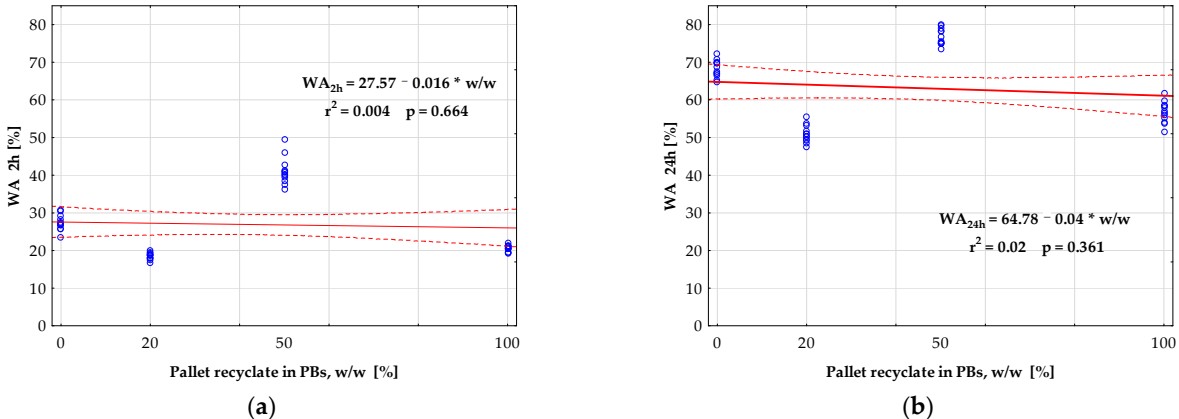

**Figure 6.** Water absorption (WA) after 2 h (**a**) and 24 h (**b**) of PBs containing different amount of particles from recycled spruce pallets (RSP).

Theoretically, no significant effect of the type of spruce particles in PBs on their TS and WA values can be explained by a similar chemical composition (e.g., amount of amorphous and crystal cellulose; type and amount of hemicelluloses, lignin and extracts) and chemical-physical characteristics (e.g., polymerization degree of cellulose; amount, space accessibility and reactivity of hydroxyl groups) of wood substance presented in the FSL and RSP. The FTIR analyses indicated that the basic chemical composition of wood particles obtained from the FSL and RSP was almost the same (Table 1). Absorbance (A) peaks of both types of spruce particles were usually very similar, i.e., the 1334 $cm^{-1}$ (specific for syringyl lignin [34]), the 1372 $cm^{-1}$ (belonging to CH deformations in cellulose and hemicelluloses), the 1430 $cm^{-1}$ (assigned to aromatic skeletal vibrations, and to C-H plane vibration in plane cellulose [35], the 1510 $cm^{-1}$ (C=C stretching of the aromatic skeletal vibrations in lignin [36]), and the 1653 $cm^{-1}$ (belonging to conjugated carbonyls -C=C-C=O [37]). Subsequently, by comparing the absorbance ratios in the FSP and RSP particles, the crystallinity indexes of cellulose [38], i.e., the "TCI" (total crystallinity index = A1372/A2900 = 0.69–0.71) and also the "LOI" (lateral order index, or empirical crystallinity index = A1430/A898 = 1.06–1.09), were almost identical for both types of spruce particles. On the contrary, the particles from RSP had a lower absorbance at 1274 $cm^{-1}$, about 24.4% (specific for quaiacyl lignin), and at 1600 $cm^{-1}$, about 32% (belonging to aromatic skeletal vibration in lignin, and –C=O stretching [34]), as well as a partly higher absorbance at 1730 $cm^{-1}$, about 17.4% (stretch of unconjugated –C=O groups of aldehydes, ketones, carboxylic acids and esters in lignin and hemicelluloses [36,39]). Differences in these three A peaks

(1274, 1600, and 1730 cm⁻¹) can be attributed to more factors in recycled pallets, e.g., to a local white-rot and a long-term atmospheric oxidation.

Achieved results point to a similar chemical structure in both types of spruce particles that cannot be generalized, because in practice the wood structure (chemical, anatomical, geometry) of fresh logs and similarly of recycled pallets can be deteriorated differently and into various degrees by biological pests (decaying fungi, insects, etc.) and abiotic effects (fire, UV irradiation, etc.). It also can be observed that the degradation processes in wood sometimes has an undefinable effect on its moisture properties.

No significant effect of particles from the recycled spruce wood (RSP) on the moisture properties (TS and WA) of tested PBs could, in theory, be explained by a similar chemical composition (cellulose, hemicelluloses, lignin, extracts) and physical–chemical properties (e.g., amount, accessibility and reactivity of hydroxyl groups) of the fresh and older spruce woods used in the experiment. However, these results and evaluations cannot be general, because in practice the molecular and anatomical structure of spruce wood in fresh logs and similarly in recycled pallets can be deteriorated differently and into various ranges— e.g., in forest, transport, in stock, during exposure.

The mechanical properties of PBs were negatively affected by the presence of particles from the recycled spruce pallets—i.e., comparing to the reference/control PB which contained particles only from the fresh spruce logs—the MOR decreased by about 31.5% from 14.6 MPa to 10 MPa, the MOE about 23.1% from 2616 MPa to 2012 MPa, and the IB by about 22.8% from 0.79 MPa to 0.61 MPa (Table 3, Figures 7 and 8). In the linear correlations "MOR, MOE, or IB = a + b × *w/w*", the coefficients of determination r² ranged from 0.43 to 0.63 (Figures 7 and 8) and p-values were 0.000, confirming a significant negative effect of particles from the old spruce pallets on the searched mechanical properties of PBs. This result could be attributed to presence of some portion of deteriorated or polluted wood in PBs based on particles from the RSP, as it is well known that wood attacked by decaying fungi or aggressive chemicals has lower strength, elasticity and hardness [40,41]. Searched mechanical properties of samples or slabs from RSP were lower by 10% ($\sigma_{Compression\parallel}$) and by 18% ($MOE_d$) than those from FSL (see point Section 2.1.1).

Generally, results related to lower mechanical properties of PBs containing particles from recycled spruce pallets are in accordance with [42–46]. Compared to the requirements by the standard EN 312 [33], the PBs based on RSP achieved in all cases the IB 0.35 MPa needed for the particleboard type P2.

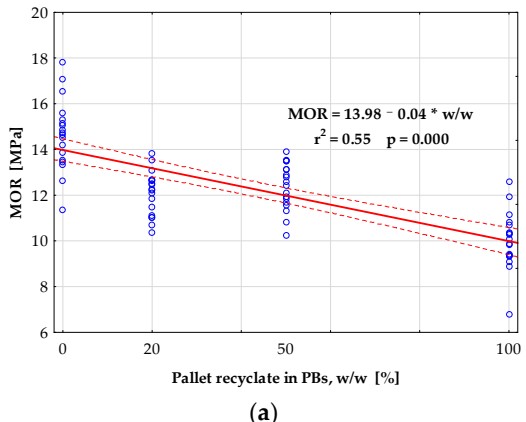

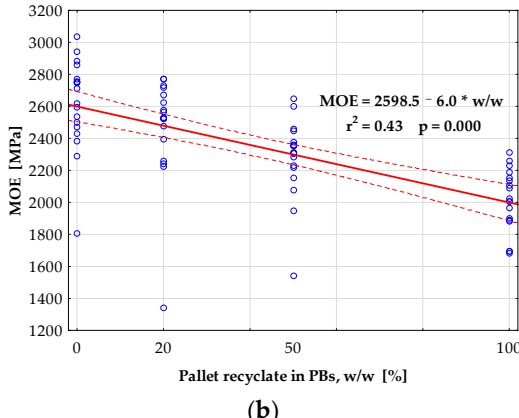

(**a**)                                                      (**b**)

**Figure 7.** Modulus of rupture (MOR) (**a**) and modulus of elasticity (MOE) (**b**) of PBs containing particles from recycled spruce pallets (RSP).

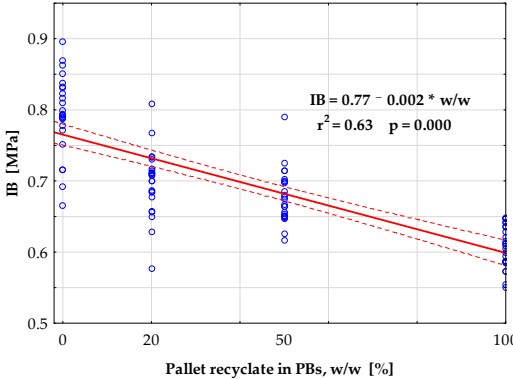

**Figure 8.** Internal bond (IB) of PBs containing different amount of particles from recycled spruce pallets (RSP).

### 3.2. Biological Resistance of PBs

The decay attack of PBs with the brown-rot fungus *Serpula lacrymans* was only slight, with a maximum of about 15.4%—inhibited at a higher portion of particles from RSP (Table 4, Figure 9). Due to the 16-week action of *S. lacrymans*, the average mass loss ($\Delta$m) of the reference PB-C was equal to 13.20%, while the PB-RSP-100 (containing only particles from RSP) was slightly higher equal to 15.23%. The coefficient of determination $r^2$ of the linear correlation "Mass loss = a + b × $w/w$" was 0.325, and the *p*-value was 0.002. These parameters confirmed only with moderate significance a higher intense rot process in those PBs which contained particles from recycled pallets (Figure 9a). After decay tests, the highest moisture content ($w_{decayed}$) had the PB-RSP-100 with the maximum portion of particles from RSP. It indirectly pointed to their more pronounced accessibility for rot (Figure 9b). The performed decay tests seemed to be valid because the strain of *S. lacrymans* caused 26.80% mass loss in the solid *Pinus sylvestris* sapwood, i.e., more than 20% required by the standard EN 113 [47].

**Table 4.** Biological resistance of the reference/control PB (PB-C) and the PBs containing particles from recycled spruce pallets (PB-RSP): (I) decay resistance valued on the basis of mass loss ($\Delta$m) caused by *S. lacrymans*; (II) mould resistance to against a mixture of microscopic fungi valued on the basis of mould growth activity (MGA) on the top surface of PBs (MGA from 0 to 4).

| Biological Resistance of PB | Recycled Spruce Pallets (RSP) in PB $w/w$ (%) | | | |
|---|---|---|---|---|
| | 0 | 20 | 50 | 100 |
| Decay attack by S. lacrymans | | | | |
| $\Delta$m [%] | 13.20 (0.48) | 12.34 (0.56) | 12.77 (0.66) | 15.23 (1.61) |
| w [%] | 87.29 (7.23) | 89.94 (2.18) | 93.41 (5.28) | 114.11 (5.85) |
| Attack by mixture of moulds (MGA [0–4]) | | | | |
| 7th day | 1.33 | 1 | 1 | 1 |
| 14th day | 2.33 | 2 | 2 | 2 |
| 21st day | 2.67 | 2.33 | 2.33 | 3 |
| 28th day | 4 | 3.67 | 3.67 | 4 |

Notes: Mean values of mass loss ($\Delta$m) in PB caused by *S. lacrymans*, as well as of mould growth activity (MGA) on the top surface of PB are from 6 samples. Standard deviations are in the parentheses.

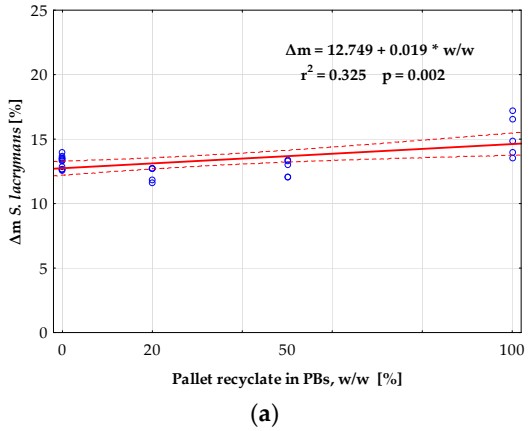
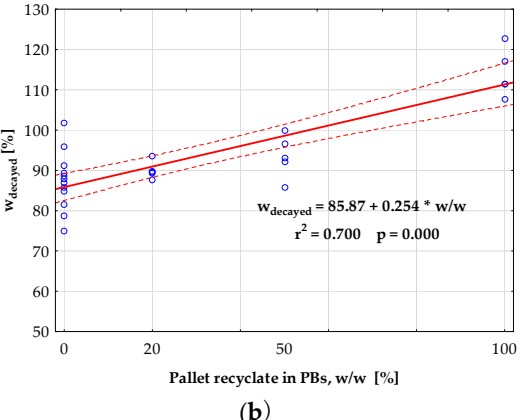

| (a) | (b) |

**Figure 9.** Mass loss Δm) (**a**) and moisture content (w$_{decayed}$) (**b**) of PBs containing different amount of particles from recycled spruce pallets (RSP) after attack by the brown-rot fungus *S. lacrymans*.

As mentioned above, spruce wood used for pallets can be deteriorated differently during storage and usage by several biological pests, such as decaying fungi, microscopic fungi, bacteria, etc. [48], at which the combined biological attack of spruce wood by more hts is usually more intensive; however, in Ref. [49], when spruce samples were primary intentionally damaged by one of the brown-rot fungi *Serpula lacrymans*, *Coniophora puteana* or *Gloeophyllum trabeum*, and then attacked again by other fungus of the aforementioned varieties, it was found that these brown-rot fungi are able to degrade the sound and primary rotten spruce wood with a comparable intensity. Some moulds (microscopic fungi) and bacteria can be useful for the subsequent wood decay caused by white-rot and brown-rot fungi; for example, *Bacilus* spp. bacteria disrupts pit membranes in cell walls of wood and thus helps the hyphae pass through tissues [50]; indeed, spruce samples primarily intentionally pretreated by the bacteria *Bacilus subtilis* had a slightly lower resistance to rot caused by the brown-rot fungus *S. lacrymans* comparing to sound spruce samples [51]. Generally can be stated, that a longer storage of spruce pallets or chips before their processing into PBs is undesirable.

The mould resistance of PBs evaluated after 7, 14, 21 and 28 days is summarized in Table 4. From the achieved results, it is evident that the kinetic and maximal rating values of the mould growth activity (MGA) on the top surface of PBs was not affected by particles from RSP; on the final (28th) day after inoculation, at least one half of PB surfaces was usually covered with hyphae of moulds, and the average MGA ranged from 3.67 to 4. Generally, surfaces of less durable and chemically unprotected woods and wooden composites are easily accessible to inhabitation by various moulds, mainly in a wet environment [52–54].

## 4. Conclusions

- The thickness swelling (TS) and water absorption (WA) values of the laboratory prepared particleboards (PBs)—based on particles from fresh spruce logs (FSL) and recycled spruce pallets (RSP)—were not affected by the particle type at all.
- On the contrary, the particles from RSP had a significantly negative effect on the mechanical properties of PBs—i.e., in connection with a decrease in the modulus of rupture (MOR) in bending up to 31.5% (from 14.6 to 10.0 MPa), the modulus of elasticity (MOE) in bending up to 23.1% (from 2616 to 2012 MPa), and the internal bond (IB) up to 22.8% (from 0.79 to 0.61 MPa).
- The particles from RSP had a significantly negative effect (but a maximum of 15.4%) on the decay resistance of PBs to the brown-rot fungus *Serpula lacrymans*. On the contrary, the mould resistance of PBs was not influenced by the type of spruce particles used.

- Generally, the manufacturing of PBs with the addition of recycled wood pallets is a very important issue from an economic and environmental point of view; however, the mechanical properties of PBs prepared from recycled wood pallets could be reduced, especially in those cases where in the used wood are damages caused by pests (fungi, insects, etc.), as well as at sorting undetected additives (biocides, paints, etc.), by which the wettability, adhesion and thus also the strength of the glued joints in the PBs is worsened.
- More representative conclusions on the optimization of PBs properties can be drawn only after more extensive laboratory and field research using several raw wood material sources, e.g., fresh logs, decayed elements from old buildings, recycled pallets and furniture, recycled modified wood, and of course, their mixtures.

**Author Contributions:** Conceptualization, J.I., L.R. and Z.V.; methodology, J.I., L.R. and Z.V.; software, J.I. and Z.V.; validation, J.I., L.R. and Z.V.; formal analysis, J.I. and L.R.; investigation, J.I., L.R. and Z.V.; resources, J.I., L.R. and Z.V.; data curation, J.I., L.R. and Z.V.; writing—original draft preparation, J.I., L.R. and Z.V.; writing—review and editing, J.I. and L.R.; visualization, J.I. and Z.V.; supervision, L.R.; project administration, J.I. and L.R.; funding acquisition, L.R. All authors have read and agreed to the published version of the manuscript.

**Funding:** This work was supported by the Slovak Research and Development Agency under the contract no. APVV-17-0583.

**Institutional Review Board Statement:** Not applicable.

**Informed Consent Statement:** Not applicable.

**Data Availability Statement:** The data presented in this study are available on request from the corresponding author. The data are not publicly available due to privacy restrictions.

**Acknowledgments:** We would like to thank to the Slovak Research and Development Agency under the contract no. APVV-17-0583 for support of the research published in this article. We also thank the companies Kronospan, s.r.o., Zvolen, Slovakia and Kronospan CR, spol. s.r.o., Jihlava, Czech Republic, for a providing parts of the materials for this research.

**Conflicts of Interest:** The authors declare no conflict of interest.

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
