# Peer review of "Particleboards from Recycled Pallets"

_forests, doi:10.3390/f12111597_

Round 1
Reviewer 1 Report
The idea of this research topic is simple but original. I just considering the quality of raw materials used in this study, due to it is recycled and they do not have quality control for their RSP. So, the results in this study were insignificant.
I suggested to choose only one pic for Figure 1, or change with FSL picture.
Fig 4, the X ordinate missed the 80 %
The argument in the discussion part need more scientific point of view, the mechanism of adhesion, and adding more refferences
I am wondering the type of fungi used in this study. Usually, for decaying test, white rot fungi type (Bacidiomycetes sp) are used, do you have any comments on this.
Author Response
Reviewer 1
Comments and Suggestions for Authors
The idea of this research topic is simple but original. I just considering the quality of raw materials used in this study, due to it is recycled and they do not have quality control for their RSP. So, the results in this study were insignificant.
Response
- Of course we agree, that the results achieved in one laboratory experiment cannot be totally significant – generally valid for practice.
- This opinion was mentioned in article on lines 193-196, as well: “… results cannot be general, because in practice the molecular and anatomical structure of spruce wood in fresh logs and similarly of spruce wood in recycled pallets can be deteriorated differently and into various ranges – g., in forest, at transport, in stock, during exposure.”
Point 1
I suggested to choose only one pic for Figure 1, or change with FSL picture.
Response to point 1:
Now for Figure 1 is used one photo of view on storage of the discarded pallets in Kronospan s.r.o., Zvolen (Figure 1a), and one photo of view from the ultrasonic measurement of pallets in the longitudinal direction for the MOEd determination (Figure 1b).
Point 2
Fig 4, the X ordinate missed the 80 %
Response to point 2:
The addition of pallets was 0%, 20%, 50%, and 100%. It means, that the 80% is not needed on the X-axis in Fig 4 (but also in Figs. 8-9).
Point 3
The argument in the discussion part need more scientific point of view, the mechanism of adhesion, and adding more references
Response to point 3:
For the moisture and strength properties of PBs are important several types of chemical bonds (etheric and other types of covalent bonds created between the wood components and the UF resin) and physico-chemical interactions (hydrogen bonds and van der Walls interactions created in the mixture of wood substrate and the UF resin) by which are connected wood particles in the PB.
Were added 2 references to adhesion in PBs.
Point 4
I am wondering the type of fungi used in this study. Usually, for decaying test, white rot fungi type (Bacidiomycetes sp) are used, do you have any comments on this.
Response to point 4:
- In this experiment two types of fungi were intentionally used: A/ For the decay test the dry rot brown-rot fungus Serpula lacrymans (it belong to the Basidiomycota phylum), which is able in the North and Central Europe, USA, etc. seriously damage products from less durable wood species, wooden materials (PBs, plywood, …) and other materials based on cellulose, typically in building interiors (kitchens, bathrooms, …); B/ For the mould test mixture of 5 microscopic fungi which are able to growth on surfaces of various organic materials, including PBs.
- White-rot fungi are able to attack wooden products mainly in exteriors. For example, in our previous experiments the chemically preserved as well as the structurally modified plywood materials recommended for outdoor expositions were tested against the with-rot fungi Trametes versicolor and Phanerochaete chrysosporium. (REINPRECHT L., KMEŤOVÁ, L., IŽDINSKÝ, J. (2012): Fungal decay and bending properties of beech plywood overlaid with tropical veneers. Journal of Tropical Forest Science 24(4): 490-497. REINPRECHT, L., KMEŤOVÁ, L. (2014): Fungal resistance and physical-mechanical properties of beech plywood having durable veneers or fungicides in surfaces. European Journal of Wood and Wood Products 72(4): 433-443.)

Reviewer 2 Report
The manuscript (ms) described particleboard (PB) making at laboratory. The PB was made by either fresh or recycle wood, and also the mixture with different proportions. Physical and mechanical properties of the PB were evaluated including its biological property, i.e. exposing to mold. Some clarifications were required in order to enhance the quality of this ms:
- Wood pallet is commonly made with nail as connector. How the authors remove this nail since it was iron or steel. This is important because the presence of iron or steel will ruin at raw material preparation, particularly when processing in flaker and grinder. These remarks were missing neither in introduction section nor materials & methods part. Authors should add this information since this study emphasized recycling.
- 3 line 96-100
- There is no information on UF resin’s basic properties. Even the adhesive was industrial/ commercial type, the authors should characterize at least their basic properties such as solid content, viscosity, pH, and gel time. All of these characterizations will help in manufacturing PB particularly in lab scale. This is mandatory to conduct!
- In my view paraffin is hydrophobic and it is used for hindering water penetrated more into PB. The paraffin used in this study was water emulsion, wasn’t it? Please add specification on this and how much the percentage when it was applied in your study. You mentioned that physical properties related to moisture (i.e. thickness swelling and water absorption) were not affected by recycle wood. In my view, role of paraffin as hydrophobic agent was significant therefore it influenced on the moist properties of the resulted PB. Further, I did not like to refer your own study except you add detail on it!
- Why there were differences on levels of adhesive used between core and surface layers? Give reason on it!
- 4 line 102-110
- Since you analyzed using ATR, why you did not also analyze the furnish (mixture wood particle and UF resin and the resulted particleboard)? If you add these data, this study will be more interesting. I am curious that hydrogen bonding participated predominantly in your PB since the moisture content of the furnish was higher compare to the wood particles.
- Again, I did not like to refer your own study (self-citation), except you give detail on them, for example type of hot pressing used (frame or column type) even the photograph.
- Are you sure that you used 240 C for hot pressing? Curing temperature of UF resin is around 160 C. I am worry that this is the reason that your PB was overbaked thus the resulted mechanical properties were lower.
- 5 line 126-136
- Again, there were self-citations! Avoid to refer your own work except there was special case!
- Criteria of mold growth is according to whom?
- Results & Discussions
- Density was below on target or what? There were no remarks on the target density of the PB.
- Line 148. “…. Had an insignificant effect……” Did you analyzed the data using anova (analysis of variance)? If so, why you did not mention statistical analysis?
- Table 3. Authors should add the standard values of thickness swelling (TS), water absorption, and mechanical properties (IB, MoE, and MoR). TS seemed exceed the standard why the authors did not state in the ms? This is important since the weakness of PB generally is in higher TS and lower screw holding power.
- IB, MoE, and MoR were lower. Why you did not combine with some SEM micrographs? Using morphology in SEM examination, you can differentiate which one the rot/ decay wood particle and their interaction with UF adhesive. Also, you can compare with the fresh ones. I think this is mandatory for strengthen your claims.
- Biologycal resistance (P.8)
- “……was significantly, ….” (line 223-224). Did you use anova? In the result (Table 4) the significantly seemed disappeared.
- How to measure the mass loss since the decay PB was moist?
- Line 260. Please add the photographs of your PB’s sample exposed to mold as your work’s evidence.

Author Response
Reviewer 2
Comments and Suggestions for Authors
The manuscript (ms) described particleboard (PB) making at laboratory. The PB was made by either fresh or recycle wood, and also the mixture with different proportions. Physical and mechanical properties of the PB were evaluated including its biological property, i.e. exposing to mold. Some clarifications were required in order to enhance the quality of this ms:
Point 1
Wood pallet is commonly made with nail as connector. How the authors remove this nail since it was iron or steel. This is important because the presence of iron or steel will ruin at raw material preparation, particularly when processing in flaker and grinder. These remarks were missing neither in introduction section nor materials & methods part. Authors should add this information since this study emphasized recycling.
Response to point 1:
- Yes, thank you for this comment. From pallets, the metal (iron, steel, ...) nails and other connector types are removed directly in the company Kronospan s.r.o. Zvolen, Slovakia, using the device external Lindner’s multi-stage processing solution powerful primary shredders combined with well-engineered separation technology (magnetic separation of ferrous metals - such as nails, scrap iron and other scrap metals) and high-precision secondary shredders and then from chips were in a knife ring flakers (Knife Ring Flakers G24, GOOS Engineering spol. s.r.o. ) prepared wood particles.
Point 2
3 line 96-100
Response to point 2:
- See answers to points 3.-5.
Point 3
There is no information on UF resin’s basic properties. Even the adhesive was industrial/ commercial type, the authors should characterize at least their basic properties such as solid content, viscosity, pH, and gel time. All of these characterizations will help in manufacturing PB particularly in lab scale. This is mandatory to conduct!
Response to point 3:
- Yes, we changed the formulations.
- Two urea-formaldehyde (UF) resin types, produced in company Kronospan, were used – (a) KRONORES CB 4005 D for the surface layers of PBs added in an amount of 11 wt.%, and (b) KRONORES CB 1637 D for the core layer of PBs added in an amount of 7 wt.%. The basic characteristics of these two UF resins were as follows: molar ratio of urea to formaldehyde constantly 1 to 1.2; solid content 65.9% and 67.4% by EN 827 (x); viscosity 76 s and 86 s using Ford cup 4 mm/20 °C by EN ISO 2431 (x); pH value 9.08 and 8.62 by EN 1245 (x); gel time 81 s and 36 s by Kronospan chloride test. The NH4NO3, used as 57 wt.% water solution, was applied as hardener of both UF resins, added to these resins in amount of 2% and 4%, respectively.
Point 4
In my view paraffin is hydrophobic and it is used for hindering water penetrated more into PB. The paraffin used in this study was water emulsion, wasn’t it? Please add specification on this and how much the percentage when it was applied in your study. You mentioned that physical properties related to moisture (i.e. thickness swelling and water absorption) were not affected by recycle wood. In my view, role of paraffin as hydrophobic agent was significant therefore it influenced on the moist properties of the resulted PB. Further, I did not like to refer your own study except you add detail on it!
Response to point 4:
- Paraffin, used as 35 wt.% water emulsion, was applied on the surface and core particles in amount of 0.6% and 0.7%, respectively.
Point 5
Why there were differences on levels of adhesive used between core and surface layers? Give reason on it!
Response to point 5:
- A higher amount of hardener in the UF resin for the core layer of PB (4%) was needed due to the technological conditions of PBs preparation – i.e., because the temperature increase needed for the UF resin curing starts later in core layer as in the surface layers of PBs.
Point 6
4 line 102-110
Response to point 6:
- See answers to points 7.-9.
Point 7
Since you analyzed using ATR, why you did not also analyze the furnish (mixture wood particle and UF resin and the resulted particleboard)? If you add these data, this study will be more interesting. I am curious that hydrogen bonding participated predominantly in your PB since the moisture content of the furnish was higher compare to the wood particles.
Response to point 7:
- Yes, such analyses could also be interesting.
- However, for the final moisture and strength properties of composites, including PBs, there are important two basic factors: a) structure and properties of the input materials; b) technological processes of the composite preparation.
- As the moisture properties and mechanical properties of prepared PBs (in a context with the standard tests performed at their moisture content of 7% ± 1%), where for the mutual connection of particles in PBs and also for the strength properties of PBs are important several types of chemical and physico-chemical interactions between individual wood particles (e.g., - etheric and other types of covalent bonds created between the wood components and the UF resin; - hydrogen bonds and van der Walls interactions created in the mixture of wood substrate and the UF resin), the effect of hydrogen bonds was by our opinion a less significant or insignificant. However, presence of lower or higher portion of hydroxyl groups in wood components and methylol groups in UF resins could significantly influenced creation of covalent (etheric, …) bonds between the wood and resin molecules.
- Yes, moisture of wood particles used for PBs preparation was lower (only 2% or 4%) as of the tested PBs (app. 7%), but it was due to the defined technological conditions of PBs preparation.
Point 8
Again, I did not like to refer your own study (self-citation), except you give detail on them, for example type of hot pressing used (frame or column type) even the photograph.
Response to point 8:
- In the laboratory condition at manufacturing of PBs the standard three stage pressing diagram was used (published in Iždinský et al. 2021).
Figure 1. Standard three stage pressing diagram in the manufacturing of particleboards (PBs).
Point 9
Are you sure that you used 240 C for hot pressing? Curing temperature of UF resin is around 160 C. I am worry that this is the reason that your PB was overbaked thus the resulted mechanical properties were lower.
Response to point 9:
- Yes, as the maximal temperature of the pressing plates in the press was 240 °C (this temperature is commonly used in industrial technologies of PBs preparation). Of course, the curing temperature of the UF resin is lower, already about 100-105 °C, and such temperature is achieved at the pressing process in the core layer of PBs at a pressing factor of 8 s/mm after app. 100 seconds.
Point 10
line 126-136
Response to point 10:
- See answers to points 11 and 12.
Point 11
Again, there were self-citations! Avoid to refer your own work except there was special case!
Response to point 11:
- So, by our opinion, some self-citations are needed either for shortening the work methodology as well as for a better comparing results between the presented work and the works made previously with other materials, technologies, etc. It we applied for the Chapters 2.2 and 2.3.
Point 12
Criteria of mold growth is according to whom?
Response to point 12:
- The mould test was performed according to EN 15457, with some modifications / changes mention already in the work Iždinský et al. 2021 (i.e., changes related to other size of samples = 50 mm x 50 mm x 16 mm, and other sterilization of samples before mould test = twice for 30 min by UV light radiator).
Point 13
Results & Discussions
Response to point 13:
- See answers to points 14.-21
Point 14
Density was below on target or what? There were no remarks on the target density of the PB.
Response to point 14:
- The density of the laboratory prepared PBs was standard and comparable with density of commercial PBs, at which it was not affected by different sources of spruce wood from which the particles for the core and surface layers were prepared. It means, finally the average density of all four PB-types ranged in a very narrow interval from 651 kg.m-3 to 657 kg.m-3 (Table 3, Figure 4).
Point 15
Line 148. “…. Had an insignificant effect……” Did you analyzed the data using anova (analysis of variance)? If so, why you did not mention statistical analysis?
Response to point 15:
- We corrected.
- … Had only a negligible impact on the moisture properties of PBs, … . It is evident from the linear correlations “TS … of PBs, which were characterized … .
Point 16
Table 3. Authors should add the standard values of thickness swelling (TS), water absorption, and mechanical properties (IB, MoE, and MoR). TS seemed exceed the standard why the authors did not state in the ms? This is important since the weakness of PB generally is in higher TS and lower screw holding power.
Response to point 16:
- The minimal limit values for the individual tested properties of commercially prepared PBs type P2 Requirements for boards for interior fitments (including furniture) for use in dry conditions (EN 312) were: MOR – 11 MPa; MOE – 1600 MPa; IB – 0.35 MPa.
Point 17
IB, MOE, and MOR were lower. Why you did not combine with some SEM micrographs? Using morphology in SEM examination, you can differentiate which one the rot/ decay wood particle and their interaction with UF adhesive. Also, you can compare with the fresh ones. I think this is mandatory for strengthen your claims.
Response to point 17:
- Yes, of course, we agree with your opinion and recommendation. Wood attacked by brown-rot, but also by soft-rot, has a lower portion of polysaccharides comparing to the fresh healthy wood – and therefore portion of its polar hydroxyl groups needed for wetting and adhesion processes with polar UF resins is a lower. In this fact the strength of PBs can also be lower.
- By our experience, SEM micrograph analyse give mainly qualitative results, as it can be influenced by selection of non-homogenous micro-samples from recycled wood or PBs. For example, our experiments with 2, 4, 8, 16, 32 and 64 weeks rotting of more wood species using more white- and brown-rot fungi shoved that changes in the anatomy of wood cells were typical for the type of rot and even the species of fungus, however, while the time-factor (= degree of mass loss) was a less representative due to non-homogenous action of rot in wood samples. (REINPRECHT, L., LEHÁROVÁ, J. (1997): I.. Microscopic analyses of beech (Fagus sylvatica) wood in different stages of rot by fungi Serpula lacrymans, Coriolus versicolor and Schizophyllum commune. II. Microscopic analyses of fir (Abies alba Mill.) and spruce (Picea abies L. Karst.) wood in different stages of rot by fungi Serpula lacrymans, Coriolus versicolor and Schizophyllum commune. In: Drevoznehodnocujúce huby ´97, Sympózium, TU Zvolen, Slovakia, 1997, p. 91-101; p. 103-113.
Point 18
Biologycal resistance (P.8)
Response to point 18:
- See answers to points 19.-21.
Point 19
“……was significantly, ….” (line 223-224). Did you use anova? In the result (Table 4) the significantly seemed disappeared.
Response to point 19:
- So, by the linear correlation only the positive or negative tendencies we wanted to determine (similarly as for physical and mechanical properties of PBs) – so for these targets we did not use Anova.
- The sentences in lines 223-224 and 232-234 were corrected.
Point 20
How to measure the mass loss since the decay PB was moist?
Response to point 20:
- The mass losses were computed for each PB sample from its weight values measured in the oven dry state before (mo) and after decay process by lacrymans in Kolle flasks lasting 16 weeks (mo/Fungally-Attacked/ (see Iždinský et al. 2021):
|
|
Point 21
Line 260. Please add the photographs of your PB’s sample exposed to mold as your work’s evidence.
Response to point 21:
- In this study the photos from mould tests were not done, as differences were minimal.

Reviewer 3 Report
Title:
Particleboards from recycled pallets
Author: Iždinský et al.
General
The article describes the analysis of the mechanical and chemical properties as well as the resistance to mould of particleboards made from fresh and recycled wood. The different ratios between fresh and recycled wood were analysed, with the greatest influence of the recycled wood on the mechanical properties such as the modulus of elasticity, the modulus of rupture and internal bond. The values of the above properties decrease with the use of recycled wood. The article is very nicely and thoroughly written.
General comments
In spite of nicely and thoroughly written study, it has one important flaw. The study does not include input data for fresh and, more importantly, recycled wood. The results only show that the modulus of elasticity, modulus of rupture and internal bonding decrease with the proportion of recycled wood, with the maximum decrease in properties for particleboard made entirely from recycled wood. In the latter, the modulus of elasticity, modulus of rupture and internal bonding decrease by 23.1%, 31.5% and 22.8%, respectively, compared to particleboards made from fresh wood.
Does this mean that recycled wood has lower mechanical properties at the same percentage, or that the mechanical properties were even lower, with smaller differences in particleboard because most of the loading was taken by glue or particles impregnated with glue?
What are the mechanical properties of recycled wood? Was the wood completely decomposed? What would be the mechanical properties of the particleboard if the recycled wood was intact?
For these reasons, it is necessary that the authors indicate the properties of the recycled material or at least the mechanical part!
I also suggest that the authors give the mechanical properties of fresh wood.
It is also not clear from the conclusions in which ratio the use of recycled and fresh wood is reasonable!
I suggest that the authors indicate the extent to which it is useful to use recycled wood in terms of the degree of decay or the values of mechanical properties of the recycled wood regarding to the required mechanical properties of the particleboard.
Only such a research will have a practical use which is now not apparent.
Specific comments
In the description of the statistical analysis (L139 -140), linear correlations with the coefficient of determination will be made. In addition to the above, p value is also used in individual results. What does the p value mean?
Author Response
Reviewer 3
Comments and Suggestions for Authors
Title: Particleboards from recycled pallets
Author: Iždinský et al.
General
The article describes the analysis of the mechanical and chemical properties as well as the resistance to mould of particleboards made from fresh and recycled wood. The different ratios between fresh and recycled wood were analysed, with the greatest influence of the recycled wood on the mechanical properties such as the modulus of elasticity, the modulus of rupture and internal bond. The values of the above properties decrease with the use of recycled wood. The article is very nicely and thoroughly written.
Point 1
General comments
In spite of nicely and thoroughly written study, it has one important flaw. The study does not include input data for fresh and, more importantly, recycled wood. The results only show that the modulus of elasticity, modulus of rupture and internal bonding decrease with the proportion of recycled wood, with the maximum decrease in properties for particleboard made entirely from recycled wood. In the latter, the modulus of elasticity, modulus of rupture and internal bonding decrease by 23.1%, 31.5% and 22.8%, respectively, compared to particleboards made from fresh wood.
Response to point 1:
- Yes, input data for the fresh spruce wood versus the recycled spruce wood are important.
- The study now contains basic input information for wood from fresh spruce logs and recycled spruce pallets – compression strength parallel with grains and dynamic modulus of elasticity (see now in Chapter 2.1.1) and data from FTIR (were in Table 1).
Point 2
Does this mean that recycled wood has lower mechanical properties at the same percentage, or that the mechanical properties were even lower, with smaller differences in particleboard because most of the loading was taken by glue or particles impregnated with glue?
Response to point 2:
- The final mechanical properties of PBs result from their macro-composition (3 layer of all 4 PB types), micro-composition (species, quality, portion and distribution of wood particles, resins, additives = equal in all 4 PB types), density (very similar – see table 2), moisture content (after conditioning equal for all 4 PB types = app. 7%), and also from technological processes of their preparation (equal).
- Predominant effect on the mechanical properties of PBs has the adhesion of wood particles with UF resin, which is influenced by wood wettability. The beginning wettability of fresh wood substrate is usually changed in time, e.g., due to weathering, rot processes, contacts with oils, industrial, food and other commodities. Surface of particles prepared from the RSP (recycled spruce pallets), which history was not known, could also be changed and some information we obtained by FTIR analyses (Table 1).
Point 3
What are the mechanical properties of recycled wood? Was the wood completely decomposed? What would be the mechanical properties of the particleboard if the recycled wood was intact?
For these reasons, it is necessary that the authors indicate the properties of the recycled material or at least the mechanical part!
Response to point 3:
- The spruce wood of recycled pallets used for PBs was outwardly healthy and its dynamic modulus of elasticity, determined with Fakopp device (Figure 1b).
- The compression strength of spruce wood from the recycled pallets comparing to spruce wood from the fresh logs was a partly lower (app. by 10% - see Chapter 2.1.1).
Point 4
I also suggest that the authors give the mechanical properties of fresh wood.
Response to point 4:
- Yes, it was done – see answers to the points 1 and 3.
Point 5
It is also not clear from the conclusions in which ratio the use of recycled and fresh wood is reasonable!
I suggest that the authors indicate the extent to which it is useful to use recycled wood in terms of the degree of decay or the values of mechanical properties of the recycled wood regarding to the required mechanical properties of the particleboard. Only such a research will have a practical use which is now not apparent.
Response to point 5:
- In the experiment used type of spruce pellets the reasonable ratio of fresh wood to recycled wood in PBs is in the range of 50 : 50 (± 30)”.
- Required mechanical properties of PBs are dependent by their usage (EN 312) – furniture, buildings, etc. (P1 – P7). So, particles from decayed wood can be in a higher portion presented in those PB-types which are used for non-construction products.
- Yes, we agree that the practical use of presented results is not entirely apparent. However, such conclusions can be made only after more extensive field research using several raw material sources (fresh logs, recycled pallets, furniture, decayed wood from buildings, modified wood, etc., and their mixtures).
Point 6
Specific comments
In the description of the statistical analysis (L139 -140), linear correlations with the coefficient of determination will be made. In addition to the above, p value is also used in individual results. What does the p value mean?
Response to point 6:
The p value represents - probability value (p-value) that measures the likelihood that a test-statistic value could occur by chance, given that the null hypothesis is true.

Round 2
Reviewer 1 Report
Thank for the responses.
This article indicates self citation and salami-indication is not appropriate, probably they need to exclude the self citation
Author Response
Thank you for the review.
Now we have excluded three our papers from the article, with the aim to decrease the self-citations.
However, we think that the other four self-citations which remained in the article are important for the article with the aim to limit the range of methodology similarities in its “Experimental part” with the previous works (i.e., we give needed references on the previous works).
Reviewer 2 Report
The manuscript has been accepted in the present form.
Author Response
Thank you for the review.
Reviewer 3 Report
The authors have substantially corrected the manuscript.
Nevertheless, there are still some minor mistakes:
L78-80 "compression strength parallel to the grain by STN 49 0110 for 15 replicates was 42.3 MPa and 38.2 MPa, respectively"
Respectively to what - you mentioned two different values, which relates to what?
L82-85 The same situation as above.
Author Response
Thank you for the review.
As stated at the beginning of the sentence, the values refer to FSL and RSP. They are also listed in such a sequence.
We used adverb “respectively” – (generally of two or more items) with each relating to something previously mentioned, in the same order as first mentioned (i.e., FSL and RSP).